# Bridging the Gaps: Investigating the Complex Impact of the COVID-19 Pandemic on Tuberculosis Records in Brazil

**DOI:** 10.3390/tropicalmed8090454

**Published:** 2023-09-20

**Authors:** Carlos Dornels Freire de Souza, Epaminondas Ribeiro Dias Neto, Thais Silva Matos, Ana Carolina Furtado Ferreira, Márcio Bezerra-Santos, Adeilton Gonçalves da Silva Junior, Rodrigo Feliciano do Carmo

**Affiliations:** 1Department of Medicine, Federal University of Vale do São Francisco (UNIVASF), Petrolina 56304-205, Brazil; nonds@hotmail.com (E.R.D.N.); anacff94@gmail.com (A.C.F.F.); 2Department of physiotherapy, University of Pernambuco (UPE), Petrolina 56328-900, Brazil; thais.matos@upe.br; 3Department of Medicine, Federal University of Alagoas (UFAL), Arapiraca 57309-005, Brazil; marciobezerra.ufs@outlook.com; 4City Hall of Juazeiro, Bahia 48903-400, Brazil; adeiltonjunior.7@gmail.com; 5Postgraduate Program in Health and Biological Sciences, Federal University of Vale do São Francisco (UNIVASF), Petrolina 56304-205, Brazil; carmorodrigof@gmail.com; 6Postgraduate Program in Biosciences, Federal University of Vale do São Francisco (UNIVASF), Petrolina 56304-205, Brazil

**Keywords:** coronavirus, pandemic, SARS-CoV-2, spatial analysis, tuberculosis

## Abstract

Background: This study aimed to analyze the temporal evolution, spatial distribution, and impact of the COVID-19 pandemic on tuberculosis records in a northeastern state of Brazil. Methods: This is an ecological study involving all diagnoses of Tuberculosis (TB) in residents of the state of Pernambuco/Brazil. Data were extracted from the National System of Notifiable Diseases. A pre-pandemic COVID-19 temporal analysis (2001–2019), a spatial analysis before (2015–2019) and during the first two pandemic years (2020–2021), and the impact of the COVID-19 pandemic on cases of TB diagnoses in Pernambuco in the years 2020 and 2021 were performed. Inflection point regression models, Global and Local Moran’s statistics, and spatial scan statistics were used. Results: In the period from 2001 to 2019, 91,225 cases of TB were registered in Pernambuco (48.40/100,000 inhabitants), with a tendency of growth starting in 2007 (0.7% per year; *p* = 0.005). In the pre-pandemic period (2015–2019), 10.8% (n = 20) of Pernambuco municipalities had TB incidence rates below 10/100,000. In 2020, this percentage reached 27.0% (n = 50) and in 2021 it was 17.8% (n = 33). Risk clusters were identified in the eastern region of the state, with five clusters in the pre-pandemic period and in 2021 and six in 2020. In the first year of the pandemic, an 8.5% reduction in the number of new TB cases was observed. In 2021, the state showed a slight increase (1.1%) in the number of new TB cases. Conclusions: The data indicate that the COVID-19 pandemic may have caused a reduction in the number of new TB case reports in the state of Pernambuco, Brazil.

## 1. Introduction

Tuberculosis (TB) is an infectious disease caused by one of the seven species that make up the Mycobacterium tuberculosis complex [1,2]. The disease affects mainly the lungs, and its transmission occurs through droplets of saliva containing mycobacteria expelled by carriers of the active lung or laryngeal disease when they speak, cough, sing, or sneeze [3,4]. Additionally, it is one of the leading causes of death worldwide and the leading cause of death from an isolated infectious agent [4,5,6]. 

Brazil ranks 20th on the list of 22 countries with the highest burden of tuberculosis in the world and is considered a priority country by the WHO for controlling the disease [7]. Additionally, Brazil did not meet the United Nations (UN) Millennium Development Goal target of a 50% reduction in tuberculosis mortality [8]. The country also stands out for its participation in the BRICS (a bloc formed by Brazil, Russia, India, China, and South Africa), whose countries account for approximately 50% of the world’s tuberculosis cases [7].

As part of the global effort to end TB, the Brazilian Ministry of Health has drawn up a national plan called “Brazil free of tuberculosis. National plan for the end of tuberculosis as a public health problem”, which aims to reduce the incidence rate to less than 10 cases per 100,000 inhabitants by the year 2035, and to reduce the tuberculosis mortality rate to less than 1 death per 100,000 inhabitants in the same period [7]. To reach these targets, the End TB Strategy includes the establishment of three pillars: integrated tuberculosis prevention and care focused on the person with tuberculosis, bold policies and support systems, and, finally, intensified research and innovation [7].

Globally, many studies have taken a spatiotemporal approach to TB incidence, helping to identify areas at risk and priority areas for intervention, such as studies in Changsha [8] and Iran [9]. These studies point to the need to expand health services in areas at high risk of transmission.

However, the world was surprised by the emergence of a new and challenging disease, of which first cases were reported as recently as December 2019. On 7 January 2020, Chinese authorities confirmed the discovery of a new type of coronavirus in Wuhan city, Hubei province, in the People’s Republic of China [10]. The etiologic agent was named SARS-COV-2 and the disease caused by it was named COVID-19 (coronavirus disease 2019) [10]. On 30 January 2020, the outbreak of the new coronavirus was declared a Public Health Emergency of International Concern (PEMI) by the WHO, which is the organization’s highest level of alert, and on 11 March 2020, it was declared a pandemic [11]. In Brazil, the first confirmed case of the new coronavirus was diagnosed on 26 February 2020 [12].

The COVID-19 pandemic was a health crisis that directly impacted the economy and the income generation of the population since restrictive measures were adopted for contingency in order to prevent the transmission of COVID-19 in most countries [13]. The world’s Gross Domestic Product (GDP) shrank by 3.5% in 2020 [14]. The number of people living in extreme poverty (earning less than USD 1.90 per day) was 6.6% in 2019, which rose to 7.8% in 2020 [15]. Countries were affected differently, with the poorest ones being affected more intensely by a worsening of the social determinants of health that were pre-existing to the pandemic [16]. The pandemic has penalized the most vulnerable in the most intense way, impacting health, economy, and society [15].

Other infectious diseases also suffered a reduction in their detection rates during the pandemic period, such as leprosy, which reduced the number of cases by 41.4% in the year 2020 compared to the average number of cases in the last five years (2015 to 2019) in Brazil [17]. In cases of Hepatitis C, there was a 61.4% reduction in the number of diagnoses and a 62.0% reduction in the incidence rate; 14,700 individuals were not diagnosed in 2020 in Brazil [18]. In terms of TB records, a Brazilian study showed an 8.3% reduction in the first year of the COVID-19 pandemic (91,225 TB cases were expected and 83,678 were reported) [19].

This study puts forth new knowledge because it presents, in a single study, the temporal evolution, spatial distribution, and possible impact of the COVID-19 pandemic on the tuberculosis records in an area considered to have a high transmission of the disease in the country. The results can contribute to the development of strategic plans capable of mitigating the harmful effects of the COVID-19 pandemic in the fight against this disease.

Thus, this work aimed to analyze the temporal evolution, spatial distribution, and impact of the COVID-19 pandemic on tuberculosis records in Pernambuco, northeastern Brazil.

## 2. Materials and Methods

### 2.1. Study Design and Area

This is an ecological study involving all diagnoses of tuberculosis in residents of the state of Pernambuco. Pernambuco is a Brazilian state located in the northeastern region, with a territorial area of about 98,000 square kilometers, and their capital is the city of Recife [20]. According to the 2021 Brazilian demographic census, it has an estimated population of 9.674 million. According to the data from the Brazilian Institute of Geography and Statistics (IBGE), the state ranks 23rd among the 27 states in relation to per capita income and 19th when analyzing the Human Development Index (HDI) [20]. According to the Social Vulnerability Index (SVI), which uses 16 vulnerability indicators, such as illiteracy, unemployment, mortality, and child labor, Pernambuco had a score of 0.351 in 2021, which represents medium social vulnerability, placing it in the 25th position among the 27 Brazilian states [21]. 

Pernambuco has 184 municipalities along with the island of Fernando de Noronha, which are distributed in 12 Regional Health Management Areas (GERES, acronym in Portuguese) for administrative and management purposes, acting in a more localized manner in Primary Health Care (PHC) and in actions to combat endemic and neglected diseases [22] (Figure 1).

### 2.2. Study Population and Data Source

This study included all cases of tuberculosis diagnosed in the residents of the study area. The data were extracted from the Sistema Nacional de Agravos de Notificação (SINAN). SINAN is a health information system that aims to standardize the collection and processing of data on notifiable diseases and complaints in Brazil [23]. It was developed in the 1990’s and its information contributes to the analysis of the morbidity profile, impacting the definition of priority actions at the municipal, state, and federal levels [24]. The diseases and illnesses notified in SINAN are defined by the National Compulsory Notification List, and among the compulsorily notifiable diseases is Tuberculosis [25]. As recommended by the Ministry of Health, we included those records reported as “new case”, “don’t know”, and “after death”. From the absolute number of cases, the incidence rate was calculated using the following equation:

Equation (1):(1)$$Incidence\;rate\;of\;Tuberculosis)\left(TB\right)=\frac{Number\;of\;TB\;cases\;(all\;types)\;in\;place\;and\;year}{Population\;living\;in\;the\;place\;and\;year}\times 100,000$$

Additionally, the population data needed to calculate the incidence were obtained from IBGE. 

### 2.3. Study Steps

As it is a study with a triple approach (temporal, spatial, and impact of the COVID-19 pandemic), this study was divided into three phases:

#### 2.3.1. Phase 1—Pre-Pandemic Time Series Analysis

For the temporal approach, a time series of 19 years (2001 to 2019) was adopted to understand this temporal behavior in the period prior to the COVID-19 pandemic. At this stage, the years 2020 and 2021 were not included as they would represent a bias in the temporal trend due to the pandemic. Additionally, the trend was analyzed on three scales: state, health regions, and municipalities.

To perform the temporal analysis, we used the joinpoint regression model of four rates at different spatial levels: state, health regions, and municipalities. This model tests whether a line with multiple segments is statistically better at describing the temporal evolution of the data than a straight line or one with fewer segments. Thus, the model allows us to identify the temporal behavior of the indicator (whether stationary, increasing, or decreasing) through the slope of the regression line, the points where there is a change in this trend (joins), the annual percent change (APC, Annual Percent Change), and the average change for each period (AAPC, Average Annual Percent Change) [26].

Parameters used in the joinpoint analysis: minimum: 0; maximum: 4; model selection: test with 4499 permutations, 5% significance, 95% confidence interval, and date-based autocorrelation of errors. These analyses were performed in the Joinpoint Regression software, version 4.5.0.1 (National Cancer Institute, USA).

#### 2.3.2. Phase 2—Spatial Dynamics of Tuberculosis before (2015–2019) and during the COVID-19 Pandemic (2020 and 2021)

For the spatial approach, the period 2015–2019 was considered. This period was justified for two reasons: to reflect the recent spatial epidemiological dynamics and to be used to calculate the expected value for the pandemic years.

In this step, the global and local Moran’s statistic and the spatial scan statistic were used. The Moran’s statistic allows for the analysis of the spatial dependence of the data. The global Moran index is obtained from the product of the deviations of the global mean. Values can range from −1 to +1, where values close to −1 indicate a negative spatial autocorrelation, values close to +1 indicate a positive spatial autocorrelation, and values close to zero indicate an absence of autocorrelation. The model is validated by applying the pseudosignificance test [27].

Once the global autocorrelation is verified, a local spatial analysis (LISA—Local Index of Spatial Association) is performed to quantify the degree of spatial association; each location of the sample set is subjected to this analysis according to a neighborhood model, allowing for the inference of the local patterns of the spatial distribution of the variables analyzed. The Local I Moran’s is a decomposition of the Global I Moran’s, with which it is possible to develop an analysis of the local pattern of spatial data [28]. 

This model is able to indicate areas in which there is a tendency to find similar values. Each area receives a significance value and is allocated to a quadrant in the Moran’s scatterplot: Q1 (positive values, positive means) and Q2 (negative values, negative means) indicate points of positive spatial association in the sense that a location has neighbors with similar values; Q3 (positive values, negative means) and Q4 (negative values, positive means) indicate points of negative spatial association in the sense that a location has neighbors with distinct values [27].

The scan statistic, on the other hand, allows for the identification of the relative risk in addition to finding the spatial cluster. The scan statistic establishes a flexible circular window on the map, positioned over each of several centroids with radius r set at 50% of the total population at risk. If the window includes the centroid of a neighbor, then the entire area of the municipality was considered included [28]. The flexibility of the window was justified by not knowing the size of the cluster a priori since the population at risk is not geographically homogeneous. 

The test to identify clusters is based on the maximum likelihood method, where the alternative hypothesis is that there is a high risk inside the window compared to outside, allowing for the quantification of this relative risk [28]. Monte Carlo simulations (999 permutations) were used to obtain the p-values. Clusters with a *p*-value < 0.05 were considered significant.

These analyses were performed with the help of the software GeoDa 1.10 (Center for Spatial Data Science, Computation Institute, The University of Chicago, Chicago, IL, USA), SaTScan (version 9.1, National Cancer Institute, Bethesda, MD, USA), and Qgis QGis (version 2.14.11, Open Source Geospatial, Foundation (OSGeo), Beaverton, OR, USA).

#### 2.3.3. Phase 3—Impact of the COVID-19 Pandemic

To quantify the impact of the COVID-19 pandemic on the number of TB diagnoses and incidence rate, the pre-pandemic period (2015–2019) was considered to calculate the expected value for 2020 and 2021. These expected values were compared with the observed values using the following equations:

Equation (2)—Impact in 2020:(2)$$Percentage\;change\;\left(P-Score\right)=\frac{no.\;of\;confirmed\;cases\;\left(2020\right)-no.\;of\;expected\;cases\;\left(mean\;2015-2019\right)}{no.\;of\;expected\;cases\;\left(mean\;2015-2019\right)}\times 100$$

Equation (3)—Impact in 2021:(3)$$Percentage\;change\;(P-Score)=\frac{no.\;of\;confirmed\;cases\;\left(2021\right)-no.\;of\;expected\;cases\;(mean\;2015-2019)}{no.\;of\;expected\;cases\;(mean\;2015-2019)}\times 100$$
where, first, the event analyzed is the number of confirmed cases of tuberculosis; second, the expected value for the year is calculated considering the last five years prior to the beginning of the pandemic, as recommended. To quantify the impact on the incidence rate, the equations were adjusted by replacing the “number of cases” with the “incidence rate”. This model has been used in other studies [17].

The results were presented in absolute numbers and proportions. Additionally, Fridman’s non-parametric test and Conover’s post hoc test were used for the comparison of the incidence rate before (2015–2019) and after the emergence of COVID-19 (2020 and 2021).

In this step, three spatial scales were analyzed: impact at the state level, impact at the level of health regions, and impact at the municipal level. In the latter case, the following stratification was adopted in order to minimize the effects of random registration fluctuation due to the small size of most municipalities in the state: i. reduction in diagnosis: <−10%; ii. no change in diagnosis: −10 to +10%; iii. increase in diagnosis: >+10%.

These analyses were performed using the software JASP (version 0.16.1.0, University of Amsterdam/Amsterdam, The Netherlands) and Qgis QGis (version 2.14.11, Open Source Geospatial, Foundation (OSGeo), Beaverton, OR, USA).

### 2.4. Ethical Considerations

In this study, only secondary data from public domain information systems were used, in which it is not possible to identify the individuals. For this reason, no authorization from the Research Ethics Committee was necessary.

## 3. Results

### 3.1. Time Trend of Tuberculosis Incidence in Pernambuco in the Pre-Pandemic Period, 2001–2019

In the period from 2001 to 2019, 91,225 cases of tuberculosis were reported in Pernambuco, with a rate of 48.40 cases per 100,000 inhabitants. Two inflection points and three trends were observed in the time series: a growth trend of the tuberculosis incidence rate between 2001 and 2004 (APC 4.5%; *p* = 0.027), followed by a stationary pattern in the period from 2004 to 2007 (*p* = 0.182), and a return to the growth trend from 2007 onwards (0.7% per year; *p* = 0.005) (Figure 2A).

Among the municipalities, 9.73% (n = 18) showed an upward trend, with a median of 5.1 (IQR 3.4 to 16.1), especially Tracunhaém (APC 42.3%; *p* = 0.026) and Calçados (APC 38.2%; *p* = 0.008). On the other hand, 7.57% (n = 14) showed a declining trend, with a median of −3.8 (IQR −6.3 to −2.1), especially in the municipalities of Santa Terezinha (APC −26.1%; *p* = 0.021) and Flores (APC −18.5; *p* = 0.029) (Figure 2B,C).

### 3.2. Spatial Analysis of Tuberculosis before and during the COVID-19 Pandemic

In the pre-pandemic period (2015–2019), 10.8% (n = 20) of Pernambuco municipalities had tuberculosis incidence rates below 10/100,000. In the first year of the pandemic (2020), this percentage reached 27.0% (n = 50). In the second year of the pandemic, this percentage was 17.8% (n = 33). On the other hand, the number of municipalities with high incidence (50 to 100/100,000) decreased in 2020 (5.4%) and increased in 2021 (12.4%) (Figure 3A–C).

Moran’s statistics showed spatial dependence of the tuberculosis incidence rate in both the pre-pandemic and pandemic periods. Municipalities in the eastern region of the state were classified in the Q1 (high–high) quadrant, with ten in the pre-pandemic period, eight in 2020, and nine in 2021 (Figure 3 D–F). Additionally, the scan statistics confirmed the existence of risk clusters in the eastern region of the state, with five clusters in both the pre-pandemic period and in 2021, and six clusters in 2020 (Table 1). The smaller number of clusters in the figure is due to overlapping clusters (Figure 3 G–I).

### 3.3. Impact of the COVID-19 Pandemic on TB Diagnoses in the Years 2020 and 2021

In the first year of the COVID-19 pandemic (2020), we observed an 8.5% reduction in the number of new cases of tuberculosis in Pernambuco, representing the non-diagnosis of at least 392 patients and an 8.73% reduction in the incidence rate. 

At the municipal level, 57.3% (n = 106) of municipalities showed a reduction in the number of TB diagnoses when compared to what was expected for the year 2020 (Figure 4A). In 2021, this number of municipalities reduced to 42.7% (n = 79) (Figure 4B). Additionally, it was observed that the density curves of the percentage of change shifted from 2020 to 2021 (Figure 4C,D). Additionally, the Friedman test showed no difference between the expected incidence (2015–2019) and that observed in 2021 (*p* = 0.121), signaling the recovery of the incidence rate to pre-pandemic values (Figure 4E).

## 4. Discussion

The results of this study indicated a discrete growth trend in the tuberculosis records between 2001 and 2019. In 2020, the pandemic reduced the number of diagnoses by 8.5%, with an unequal impact among health regions and municipalities. In 2021, the state showed signs of recovery in the number of diagnoses, although insufficient to compensate for the under-registration observed in the previous year (2020).

In the 2021 global tuberculosis report, the WHO showed a global drop in people diagnosed with TB, decreasing from 7.1 million in 2019 to 5.8 million in 2020 [29], which means a decrease in diagnosis of about 25%. The situation becomes even more complex because TB is classified as a disease of neglected populations, along with other diseases, such as leprosy and leishmaniasis [30]. These populations already suffer from barriers to accessing health services for the diagnosis and treatment of these diseases, which has been aggravated by the pandemic context itself [13,17,31]. 

TB has been the subject of numerous spatiotemporal studies worldwide. In Changsha, Xie et al. [8] identified spatial clusters of risk from 2013 to 2016. Similar results were observed in Iran in a study by Kiani et al. [9]. These two studies, by identifying risk areas for TB transmission, indicate the need for interventions in these risk areas, such as the development of policies and strategic plans. In our study, the identification of risk areas can contribute to the control of TB in the state.

The consequences of the pandemic were recorded in different parts of the world. In Japan, the average annual number of reports of 27 infectious diseases was significantly lower in the interpandemic period (2020–2021) than in the pre-pandemic period (2015–2019), with tuberculosis standing out with a 27.5% reduction [32]. In Jiangsu province, China, tuberculosis notifications dropped by 52% in 2020 compared to 2015–2019 [33]. In Malawi, the COVID-19 pandemic caused an immediate 35.9% reduction in TB notifications in April 2020 [34]. 

In Brazil, from January to December of 2020, there was a decrease in the notification of new cases of TB of around 10.9% compared to the same period in the previous year; this reduction was greater in the month of May (−31.9%), with a decrease in all regions of the country [35]. This finding is similar to that found in our study, which showed an 8.5% decrease in the number of new TB cases in Pernambuco in 2020. All regions showed a decrease similar to that observed in the national index.

Among the factors that may explain the reduction in TB diagnosis during the COVID-19 pandemic, we can highlight: 1. the adjustments made in health services in order to reduce the risk of contamination, such as the reduction in the number of daily visits, changes in the logistics of laboratory tests, the reallocation of professionals to care for individuals with COVID-19, and the dismissal of professionals due to risk factors; 2. the population’s fear of contamination, which implied a reduction in the demand for medical care; 3. the compromise of disease surveillance actions, such as the suppression of an active search and investigation of cases [35].

Moreover, a report by the Ministry of Health indicates that the decrease in diagnosis and notification of cases were more pronounced in secondary and tertiary TB referral services (−15.8%), while in primary healthcare services (PHC), the reduction was −7.4% [35]. This scenario indicates that all levels of care were substantially affected by the COVID-19 pandemic, although primary care is the most resilient.

PHC is the main entrance door for users into the SUS in Brazil, which has essential attributes, such as the provision of first contact services, longitudinality, integrality, and care coordination [30]. Among the strategies used by the PHC in the fight against TB are home visits, BCG vaccination, active search for respiratory symptoms, and directly observed treatment [30]. 

Despite the methodological care, this study has limitations. The first concern is the impact of the pandemic itself on surveillance systems, which may result in underreporting. For this reason, the number of TB records may be slightly lower than the actual number of diagnoses. In addition, the small municipalities had their surveillance systems impacted via a reduction in the working hours, the absence of professionals to work on the front line, illness or chronic conditions, and compromising the recording of data.

## 5. Conclusions

In conclusion, the COVID-19 pandemic caused a reduction in the number of notifications of new cases of tuberculosis in the state of Pernambuco. In 2021, there was a tendency for recovery in the number of diagnoses, although this was insufficient to compensate for the underdiagnosis caused by the pandemic.

It is recommended that governments adopt public policies capable of capturing undiagnosed cases during the COVID-19 pandemic and intensify actions to address the disease at the local level in order to mitigate the damage caused by the COVID-19 pandemic.

## Figures and Tables

**Figure 1 tropicalmed-08-00454-f001:**
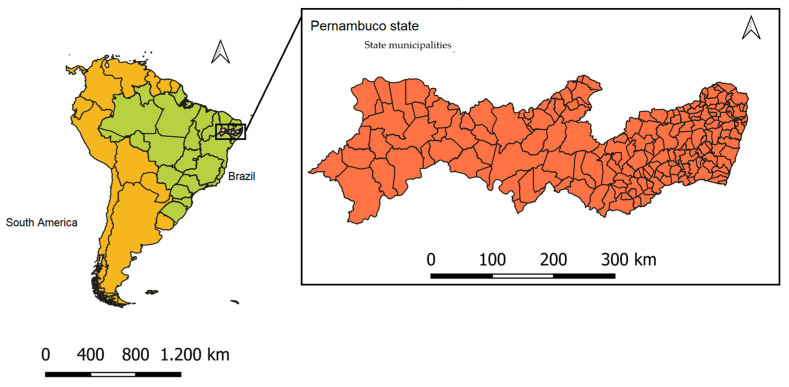
Pernambuco, Brazil.

**Figure 2 tropicalmed-08-00454-f002:**
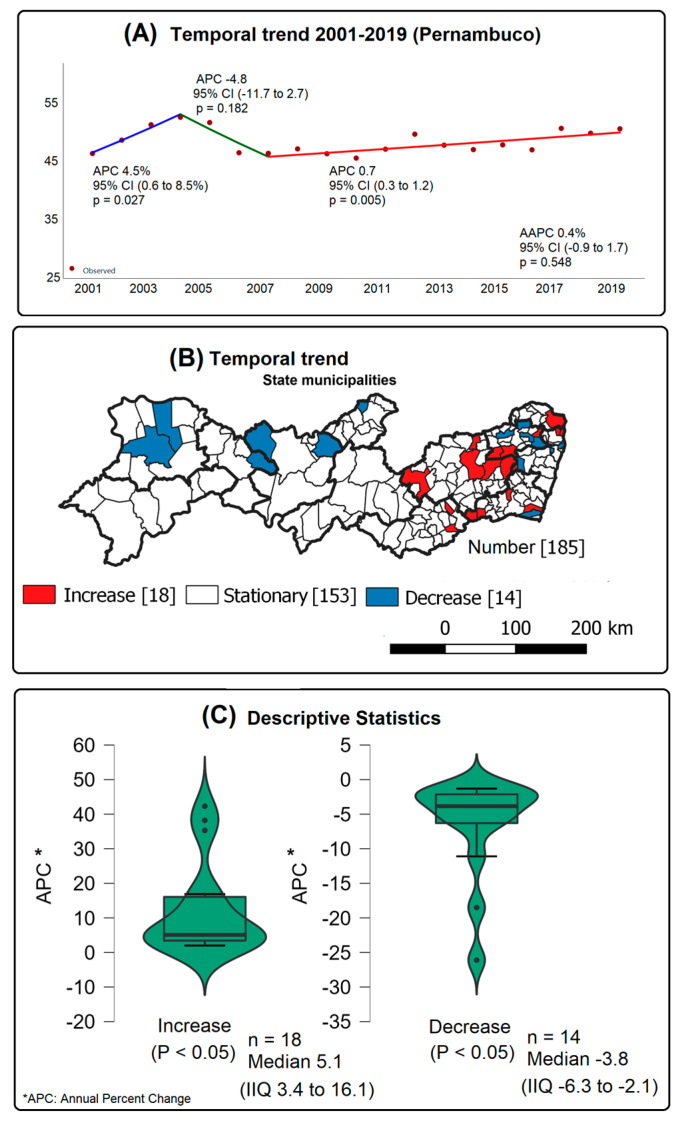
Temporal analysis of the incidence of tuberculosis in the pre-pandemic period, 2001–2019. Pernambuco, Brazil.

**Figure 3 tropicalmed-08-00454-f003:**
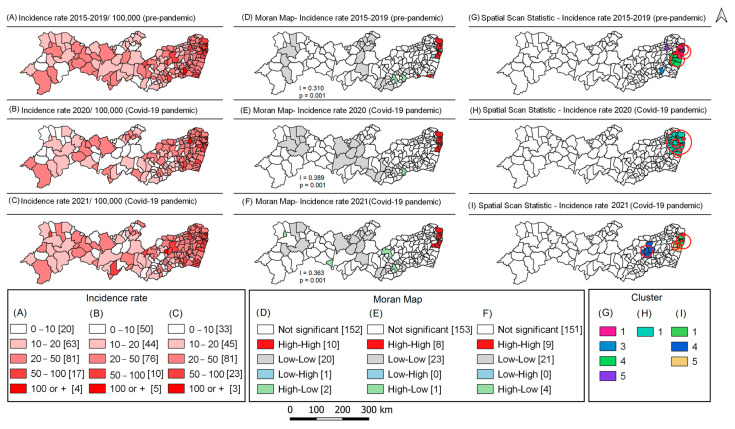
Spatial analysis of tuberculosis incidence in the pre-pandemic (2001–2019) and pandemic (2020 and 2021) periods. Pernambuco, Brazil.

**Figure 4 tropicalmed-08-00454-f004:**
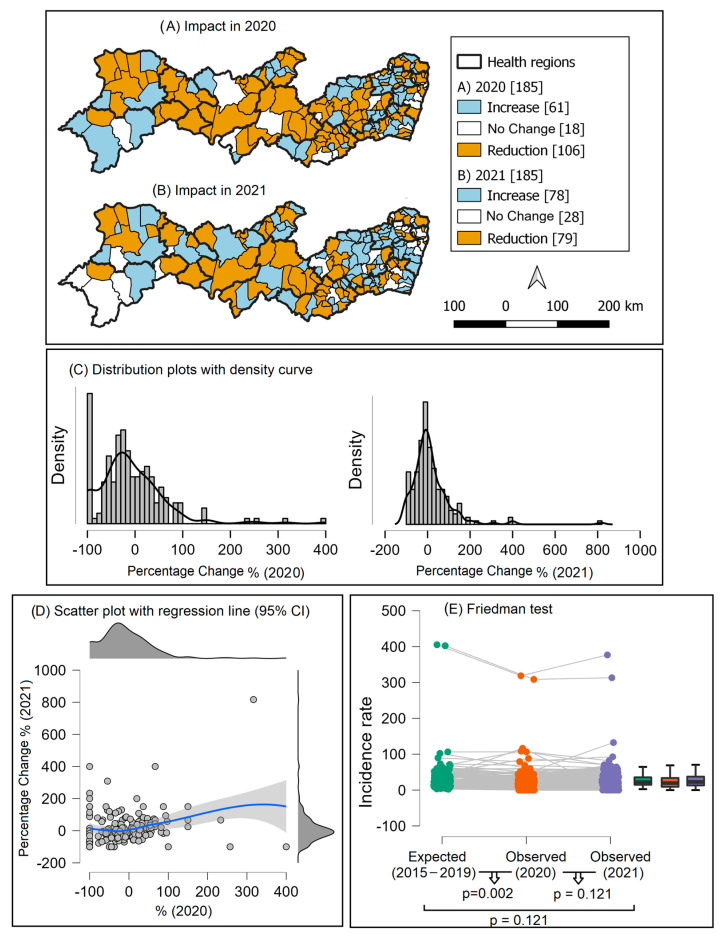
Impact of the COVID-19 pandemic on the diagnosis of new cases of tuberculosis in Pernambuco, according to municipalities in the state. Pernambuco, Brazil, 2020 and 2021.

**Table 1 tropicalmed-08-00454-t001:** Spatial scanning statistics of tuberculosis risk areas in Pernambuco, Brazil.

**(A) Pre-Pandemic Period (2015–2019)**						
Cluster	No. Municipalities	Incidence Rate	No. Cases	Radius (km)	RR	*p* Value
1	10	80.1	14,348	27.8	2.66	<0.001
2	8	85.9	11,916	20.4	2.54	<0.001
3	1	102.1	321	0.0	2.10	<0.001
4	7	53.5	3664	27.4	1.11	<0.001
5	1	71.2	201	0.0	1.45	<0.001
**(B) First year of the pandemic (2020)**						
Cluster	No. municipalities	Incidence rate	No. cases	Radius (km)	RR	*p* value
1	30	66.4	3075	47.0	2.8	<0.001
2	3	75.9	1677	9.0	2.2	<0.001
3	2	312.9	168	5.9	7.3	<0.001
4	1	108.3	61	0.0	2.5	<0.001
5	3	54.9	539	15.8	1.3	<0.001
6	7	60.9	208	15.3	1.4	0.004
**(C) Second year of the pandemic (2021)**						
Cluster	No. municipalities	Incidence rate	No. cases	Radius (km)	RR	*p* value
1	10	77.2	2827	27.8	2.5	<0.001
2	2	86.5	1782	8.9	2.3	<0.001
3	2	344.2	187	6.0	7.4	<0.001
4	4	62.7	278	22.0	1.3	0.014
5	4	56.5	623	15.8	1.2	0.030

## Data Availability

Not applicable.

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
