# Peer review of "Bridging the Gaps: Investigating the Complex Impact of the COVID-19 Pandemic on Tuberculosis Records in Brazil"

_tropicalmed, 2023, doi:10.3390/tropicalmed8090454_

Round 1

Reviewer 1 Report

This is an interesting study highlighting an important aspect of the interference of the COVID pandemic in relation to control and surveillance of other infectious diseases.

The manuscript is well written. Some aspects have to be amended, and the discussion section should be extended, as detailed in the specific comments below. 

Specific comments:

Abstract:

This is an ecological study, and while data somehow indicate that there is a link between the pandemic and registration of TB cases, a causal relationship cannot be proven. Thus, the conclusion in the abstract should be amended into “The data indicate that the pandemic of COVID-19 may have caused a reduction in the number of new TB case reports in the state of Pernambuco/Brazil“, or similar. There may be other factors (such as decreasing support of the unified health system SUS by the then Federal Government) which may have contributed to reduced detection rates.

Lines 42-45: This can be deleted, as it is common knowledge. The paper could start with “TB is one of the leading causes of death worldwide and the 46 leading cause of death from an isolated infectious agent.“ Similarly, the paragraph lines 63-72 can be deleted (with exception of the last sentence which contains specific information important for the paper). 

Methods:

Line 95: reword DESIGN AND STUDY AREA into “Study Design and Area”

There is no need to present formulas for calculation of incidence, joinpoint regression, Moran’s statistics, LISA etc. as these are standard methods described elsewhere – mentioning the method with a reference is sufficient. 

Please state clearly the definition of units of observation for the ecologoical study (the 184 municipalities, I assume). Did you exclude Fernando de Noronha? Especially for spatial analysis on the municipality level, the inclusion may cause a distortion of results, as this municipality is a small island far away in the Atlantic Ocean. If not excluded, in my opinion the analyses should be redone without Fernando de Noronha. 

Please make sure that all statistical methods are described in the methods section. For example, results of the Friedman test are presented, but the test is not mentioned in the methods section.

Discussion:

The first sentence (lines 330-331) describes the objective of the study. There is no need presenting/repeating this here – this is already placed at the end of the introduction.

The discussion should present in more detail the general impact of the COVID pandemic in Brazil, the non-responsiveness of the then Federal Government with devastating consequences not only for the population, but also for the health systems, including SUS, and (if any) the Pernambuco state government actions to mitigate the negative health effects caused by these Federal political failures.

The discussion should also be extended in more detail by the observation of similar trends for other infectious diseases (such as leprosy – which is briefly mentioned, but there is additional evidence also on other diseases).

As this is an ecological study, the authors should discuss in detail other factors that may have interfered with the TB case detection rate in recent years, such as the continuously decreasing support of the Unified Health System SUS on the federal level. What happened on the state level during this period? What about the impact of social distancing measures within the realm of TB incidence? Etc.

In general, English language is good. Some minor English errors may be corrected during typesetting.

Reviewer 2 Report

Bridging the Gaps: Investigating the Complex Impact of the COVID-19 Pandemic on Tuberculosis Records in Brazil

Specific comments:

1.     It is suggested to remove titles from the abstract. (e.g., background, etc.).

2.     It is suggested to use full words instead of abbreviations when using them for the first time (e.g., TB in the abstract section).

3.     Please check for writing errors in all the text. For example, 202012 in line 72 or 202012 in line 76.

4.     What does "City of State" mean in Figure 1 (A)? Is it a synonym for counties?

5.     I don't think there is a need to write all subtitles in uppercase. (e.g., lines 95 and 118).

·       You should review the study literature, find the gaps, and answer this question: What is your study's novelty? What methods did you use differently from others? For example, there are thousands of studies worldwide, to which you can pay attention to the most related ones so that you can include them as your study literature review sources:

·       Spatio-temporal epidemiology of the tuberculosis incidence rate in Iran 2008 to 2018

https://bmcpublichealth.biomedcentral.com/articles/10.1186/s12889-021-11157-1

·       Epidemiological characteristics and spatial-temporal clustering analysis on pulmonary tuberculosis in Changsha from 2013 to 2016

·       Impact of the COVID-19 Pandemic on the Diagnosis of Tuberculosis in Brazil: Is the WHO End TB Strategy at Risk?

6.     It is suggested that you add equation numbers to the text.

7.     Please check the spelling errors."pseudosignificance test27". In line 175 or line 197. There are several of the same issues in your text. Check for the same issues in all the texts.

8.     Which method or statistic is used to detect temporal trends? Spatial variations of temporal trends Scan Statistic is usable for this aim.

9.     Please cite the references you used. (e.g., lines 205-244). Check for the same issues.

10.  Please check which one is correct: Moran  statistics or Moran's statistic.

11.  I think this is the main but weak part of your results. "3.3. "Impact of the COVID-19 pandemic on TB diagnoses in the years 2020 and 2021". I think you should apply the best methods to show the impacts or relationships. How can we ensure that the changes are related to COVID-19? How do we prove it? Based on what evidence?

12.  What are the complex impacts of the COVID-19 pandemic on TB diagnoses, as you claimed in your title? I think this section is not complete and strong.

13.  You didn’t explore complex impacts. So, it is not true to claim this in the paper title.

14.  Based on the previous comment, you should revise the discussion part. You didn’t test any factors, but you discussed them in Lines 357–364.

15.  You should compare the study findings with other studies that have been done worldwide.

16.  What were your study's strengths and limitations?

17.  Your conclusion is short and incomplete.

Reviewer 3 Report

The study offers valuable insights into the impact of the COVID-19 pandemic on tuberculosis cases in Pernambuco, Brazil. The use of traditional statistical methods like inflection point regression models, Global and Local Moran statistics, and spatial scan statistics has provided important findings. However, given the complexity of the data and the potential for interactions among various factors, incorporating machine learning techniques could enhance the depth of your analysis. Machine learning algorithms, such as decision trees, random forests, or clustering methods, can uncover hidden patterns and relationships within the data, offering a more comprehensive understanding of the pandemic's impact on TB cases. These methods can handle nonlinear relationships and interactions that might not be captured effectively by traditional statistical approaches. Considering the ever-increasing availability of data and the growing prominence of machine learning in epidemiological research, exploring the integration of machine learning techniques could contribute to the robustness of your study's findings and further enrich the overall analysis.

English language is fine. Although, minor typographical errors have to be corrected.

Author Response

Adjusted

Round 2

Reviewer 2 Report

Thank you for your revisions. 

I think you should go back and check my previous comments carefully. For example, comment number 5, which is related to the introduction, has not been done:

·       You should review the study literature, find the gaps, and answer this question: What is your study's novelty? What methods did you use differently from others? For example, there are thousands of studies worldwide, to which you can pay attention to the most related ones so that you can include them as your study literature review sources:

·       Spatio-temporal epidemiology of the tuberculosis incidence rate in Iran 2008 to 2018

https://bmcpublichealth.biomedcentral.com/articles/10.1186/s12889-021-11157-1

·       Epidemiological characteristics and spatial-temporal clustering analysis on pulmonary tuberculosis in Changsha from 2013 to 2016

·       Impact of the COVID-19 Pandemic on the Diagnosis of Tuberculosis in Brazil: Is the WHO End TB Strategy at Risk?"

It is requested to add literature review to your introduction and revise introduction. Then you should compare your findings with literature review in discussion part. 

Author Response

Thanks for you revisions.

We adjusted the introduction and discusison sections.

Reviewer 3 Report

The study has been revised and is now acceptable for publication.

Author Response

None.